# Concrete Slab-Type Elements Strengthened with Cast-in-Place Carbon Textile Reinforced Concrete System

**DOI:** 10.3390/ma14061437

**Published:** 2021-03-16

**Authors:** Hyeong-Yeol Kim, Young-Jun You, Gum-Sung Ryu, Gi-Hong Ahn, Kyung-Taek Koh

**Affiliations:** Structural Engineering Department, Korea Institute of Civil Engineering and Building Technology (KICT), 283 Goyangdae-Ro, Ilsanseo-Gu, Goyang 10223, Korea; hykim1@kict.re.kr (H.-Y.K.); ryu0505@kict.re.kr (G.-S.R.); agh0530@kict.re.kr (G.-H.A.); ktgo@kict.re.kr (K.-T.K.)

**Keywords:** carbon textile, textile reinforced concrete (TRC), shotcrete, structural testing, flexural strengthening, bond test

## Abstract

Although carbon textile reinforcement widely used to replace the steel reinforcing bars but the bonding strength of carbon textile is generally much smaller than that of common steel bars. This study examines the strengthening effect of concrete slab-type elements strengthened in flexure by carbon textile reinforcement according to the surface coating of textile and the amount of reinforcement. The effect of the surface coating of textile on the bond strength was evaluated through a direct pullout test with four different sizes of coating material. The surface coated specimens developed bond strength approximately twice that of the uncoated specimen. The flexural strengthening effect with respect to the amount of reinforcement was investigated by a series of flexural failure tests on full-scale reinforced concrete (RC) slab specimens strengthened by textile reinforced concrete (TRC) system. The flexural failure test results revealed that the TRC system-strengthened specimens develop load-carrying capacity that is improved to at least 150% compared to the non-strengthened specimen. The strengthening performance was not significantly influenced by the textile coating and was not proportional to the amount of reinforcement when this amount was increased, owing to the change in the failure mode. The outstanding constructability afforded by TRC strengthening was verified through field applications executing TRC strengthening by shotcreting on a concrete box culvert.

## 1. Introduction

Due to lightweight, high load-bearing capacity, and high durability, textile reinforced concrete (TRC) has been widely used in new construction of structural components [1,2,3,4]. On the other hand, there are application examples of concrete structures erected by using technical textiles made of high strength fibers as the main reinforcement instead of steel bars [5,6], such TRC is mostly utilized for the strengthening of structurally deficient or deteriorated reinforced concrete (RC) structures [7,8]. As shown in Figure 1, TRC system strengthening based on textile reinforcement and a matrix is carried out directly on the surface of concrete. The textile reinforcement generally takes the form of a two-dimensional grid of carbon fibers and the matrix is commonly made of concrete or cementitious mortar, e.g., [7,8].

Since the textile generally requires a cover thickness smaller than that of the steel bars, TRC system can achieve reinforced sections that are thinner than steel reinforced concrete and presents the advantage of being possible on a wet concrete surface [8]. Unlike steel reinforcement, the carbon fiber textile does not corrode. Therefore, TRC system can be effectively applied to strengthen concrete structures built in coastal areas or environments subjected to chloride attacks or deicing agents. Recently, graphene reinforced cement composites have also been introduced in construction [9].

Previous studies examined the flexural strengthening of reinforced concrete beams [10,11,12,13,14,15,16], RC slabs [4,17,18,19], and RC bridge deck slab segment [20] by carbon TRC systems considering the grid type, the number of grid plies, the pre-tensioning of textile, and the matrix composition and strength as design variables. These studies verified that the textile made of carbon fibers, which exhibits a tensile elastic modulus much higher than glass fibers and basalt fibers, is generally effective for flexural strengthening. When the number of textile plies, that is, the amount of reinforcement, was increased, the flexural performance improved, but the stiffness and failure load enhancements were rarely proportional to the amount of reinforcement. The introduction of pre-stressing force to the textile enhanced the flexural strengthening effect compared to the non-tensioned textile. Finally, the strength of the matrix was seen to have an insignificant influence on the flexural strengthening effect.

More recently, a research group in the Korea Institute of Civil Engineering and Building Technology (KICT) conducted a series of investigations on carbon TRC system for new construction as well as strengthening for concrete elements. In KICT, a precast TRC panel was used for stay-in-place permanent formwork in new construction of reinforced concrete elements [21]. Meanwhile, slab-type concrete elements were strengthened with a precast TRC panel with grout [22] as well as a cast-in-place TRC system [23].

It is generally recognized that steel reinforcing bars develop a perfect bonding between concrete and steel. On the other hand, the bonding strength of carbon textile is generally smaller than that of common steel bars. Therefore, slip of the textile within the matrix is known as a critical behavior in TRC system strengthening [8]. There are several ways to improve the bonding performance of the textile: increasing friction, providing mechanical interlock, and increasing chemical bonding.

In the present study, the carbon textile surface was coated by an abrasive material to improve its bond performance. A direct pullout test was performed on textiles coated by a coating material with four different diameters to evaluate the effect of the surface coating on the bond strength of the carbon textile. In the aforementioned study in KICT [21,22,23], the surface of the carbon textile grid was also surface coated with an alumina oxide powder to improve the bonding performance. Moreover, the long-term mechanical performance of TRC system with surface coated carbon textile grids was investigated by two research groups in KICT and ITA RWTH Aachen University (Aachen, Germany), as a collaborative research work [24].

One of the objectives of this study is to improve the bonding performance of carbon textile grid reinforcement. In the present study, the textile surface was coated by an abrasive material to improve its friction strength. A direct pullout test was performed on textiles coated by a coating material with four different diameters to evaluate the effect of the surface coating on the bond strength of the carbon textile. Furthermore, evaluation of the flexural strengthening effect provided by the TRC system with the surface coated textile by executing flexural failure tests on RC slabs strengthened by the TRC system is another objective of this study.

Seven full-scale RC slab specimens strengthened for flexure by a carbon TRC system were fabricated and subjected to a loading test to verify the strengthening effect considering the surface coating of the textile reinforcement and the amount of reinforcement as design variables. The results of flexural failure tests for full-scale RC slab specimens with the TRC system are summarized in this paper. This paper also summarizes the TRC system strengthening procedure and results for a prototype concrete box culvert by shotcreting as a field application.

## 2. Experimental Program

### 2.1. Textile Reinforcement

Figure 2a shows the 2D carbon grid-type textile (as delivered condition, SITgrid 041 KK, Wilhelm Kneitz, Germany) used in this study. It should be noted that although textile grid shown in Figure 2a is denoted as “uncoated” but the as-delivered grid is coated with a polystyrene dispersion in the manufacturers’ factory. Table 1 summarizes the mechanical properties of the textile in the warp direction. Figure 2b depicts the textile grid with surface coating made of aluminum oxide (Al_2_O_3_) powder and vinylester resin to enhance the bond strength. Note that the aluminum oxide powder is an abrasive material used for sand papers. The grid was brushed with a resin mix consisting of vinyl ester (weight 98%), methyl ethyl ketone peroxide (weight 1%), and promoter (weight 1%). The white aluminum oxide powders were hand sprayed over the resin impregnated grid.

### 2.2. Bond Tests

Numerous research groups carried out direct pullout tests to examine the effect of textile surface coating on the bond strength [25,26,27,28]. The results of their studies indicated that the bond strength of the textile was significantly increased by surface coating. In addition, e.g., Kim et al. [23] conducted direct tensile tests on dumbbell-type specimens to investigate the effect of the textile surface coating on the tensile strength of TRC system.

The present study evaluated the effect of the size of the coating material on the bond strength. Four different sizes of coating material (Figure 3) were considered. Direct pullout tests were performed on specimens fabricated using the coated warp yarn. Figure 4 and Figure 5 respectively show the dimensions of a pullout specimen and the test setup. As shown in Figure 4, two sets of 3 mm-thick steel tabs were glued at each end of the specimen by epoxy resin to prevent failure at the grips during the loading test of the specimen. Table 2 lists the mix composition of the mortar used for the pull-out specimens. The compressive strength of the pre-mix mortar used for the mortar block was 63 MPa.

The bond strength of the textile was identified by a pullout test specified in ISO 10406 [30] and can be calculated by
(1)$$\tau =\frac{P}{u\times l}$$
where P = maximum pullout load (N); u = peripheral length of the warp (mm); and, l = embedded length (mm). Note that the embedding length used in this test is the same as the depth of the mortar block.

Table 3 summarizes the pullout test results corresponding to the considered sizes of Al_2_O_3_ powder. It appears that the specimen coated by #80 coating developed the highest bond strength and the specimen using #24 coating had the smallest coefficient of variation (CoV). All the specimens showed the interfacial failure due to slip. The influence of the diameters of the coating material on the pullout strength was insignificant because the interfacial failure was governed by the strength of the resin material used. On average, the coated specimens developed bond strength approximately twice that of the uncoated specimen. Therefore, it is verified that the textile surface coating contributes significantly to the improvement of the textile bond strength.

### 2.3. Fabrication of TRC Strengthened Specimens

A total of eight full-scale RC slab specimens were fabricated. Seven slab specimens were then strengthened for flexure at their bottom face by a TRC system with a thickness of 20 mm, as shown in Figure 6a. The remaining non-strengthened slab was used as the control specimen. Table 4 lists the characteristics of the eight slab specimens. The textile reinforcement adopted in the fabrication of the specimens was surface coated by #24 coating material since #24 coating showed the smallest CoV in the pullout test.

As shown in Figure 6b, steel reinforcement with a diameter of 16 mm was arranged in the RC slabs with three bars in the compression zone and five bars in the tensile zone. The 16 mm bars were uniformly spaced at 450 mm and 225 mm in the longitudinal direction for the top and bottom reinforcements, respectively. In addition, steel bars with a diameter of 10 mm were arranged at a spacing of 165 mm in the transverse direction. In this study, the shear capacity of the specimen is due to concrete. However, closed-type transverse rebars were placed to hold the longitudinal rebars during the fabrication and to control cracking. The compressive strength of concrete adopted for the fabrication of the specimen was 34.0 MPa by a test on cylinders at 28 days. The yield strength of the 16 mm- and 10 mm-steel bars was respectively 451 MPa and 488 MPa. The TRC system strengthening of the RC slabs was conducted in an inverted position for convenience.

Figure 7 shows a plan view of a full-scale slab specimen strengthened with the TRC system. Figure 8 illustrates the fabrication process of the specimens. As shown in Figure 8a, the surface of the RC slab was grinded 2 to 3 mm in depth to facilitate bonding with the TRC system and the top surface of the slab was maintained in a wet state for 24 h before the application of the TRC system. A steel form with a height of 20 mm was disposed on the top face of the slab to install the TRC system and the first layer of fresh mortar was laid with a thickness of about 10 mm and finished using a trowel (Figure 8b). After completion of the first mortar layer, the textile grid was put in place. When the 2nd ply of textile was placed, the meshes were crossed each other, as shown in Figure 8c. The second layer of fresh mortar was finally placed to complete the strengthening with the TRC system (Figure 8d).

Table 5 lists the mix composition of the mortar used in the TRC system. Note that we may call as textile reinforced mortar, instead of TRC, since the matrix used in this study contains no coarse aggregates. The compressive strength of mortar (air-cured at 20 °C) measured at 28 days on cubic specimens was 77.6 MPa. The purpose and process of the mortar mix design is explained in detail in a previous study [21]. Polyvinyl alcohol (PVA) fibers (KURALON K-II REC100L, Kuraray, Japan) were admixed at 1% fiber volume fraction to prevent cracking due to the drying shrinkage of the TRC system. The aforementioned study in KICT [21] indicates that incorporation of short fibers can enhance the cracking strength of TRC.

### 2.4. Test Setup and Instrumentation

A three-point bending test was conducted as shown in Figure 9 to examine the flexural performance of the slab specimens. Figure 10 illustrates the location of three foil-type strain gauges mounted on the top bar (compression, SG-3) and bottom bars (tension, SG-1 and SG-2) of the slab specimens. Note that the strain gauges were mounted to monitor the neutral axis change and yielding of steel bars. As shown in Figure 9, the vertical displacement was measured by a linear variable displacement transducer (LVDT) positioned at the center of specimen.

Load was applied to the specimen through a steel loading head (a half circle cross-section, Figure 9). Loading was applied using a universal testing machine (UTM) with capacity of 2000 kN through displacement control at a speed of 1 mm/min. Static load was applied and held after every increment of 50 kN to draw the crack maps until failure. The loading was continued until the specimen did not show a significant strain hardening behavior after the peak load.

## 3. Test Results and Discussion

### 3.1. Bonding Strength of TRC System

According to previous experimental results related to TRC system strengthened slabs [12,18], delamination may occur due to the bond failure at the interface between concrete and the TRC system. This study sought to assess the bond strength of the TRC system according to the surface treatment of RC slabs. To that end, pull-off tests [31] was carried out on the TRC system for the PO specimen (Table 4), the surface of which was grinded to a depth of 2 to 3 mm (Figure 11a) and on the same specimen where the surface was not grinded.

Table 6 lists the corresponding pull-off test results for the PO specimens. The concrete surface treatment appeared to have an insignificant effect on the bond strength, as the bond strength developed by the TRC system was much higher than the tensile strength of concrete, which has a theoretical value of 2.04 MPa. As shown in Figure 11, all the tested specimens experienced failure inside the concrete and not at the TRC system-concrete interface. However, when strengthening is performed on an existing concrete structure, the execution of grinding or chipping to remove the deteriorated surface is recommended in order to prevent delamination caused by the bond failure of the TRC system.

### 3.2. Load-Displacement Behavior

Figure 12a shows the load-displacement curves of the control slab specimen and Figure 12b–d show those of the TRC strengthened specimens. The RC specimen exhibits typical behavior starting linearly until the initiation of cracks, followed by a second stage resisting the load until yield of the steel reinforcement and, finally, a third stage showing a plateau where only the displacement increases after yield of the steel reinforcement (Figure 12a). As shown Figure 12b–d, the load-displacement curves for the specimens strengthened with the TRC system indicate four-stage behavior regardless of the surface coating of the textile grid and the amount of reinforcement. Linear behavior can be distinguished until the initiation of cracks in the first stage. In the second stage, the load is increased until the yield of steel reinforcement. The third stage shows an increase of the load followed by abrupt failure in the fourth stage. The abrupt failure of the strengthened specimens was due to the rupture of the textile reinforcement or shear failure of concrete.

By comparing Figure 12a with Figure 12b–d, it is clear that the TRC strengthened specimens induce smaller displacement than the unstrengthened one under the same load level.

### 3.3. Load-Carrying Capacity

Table 7 summarizes the flexural failure test results for unstrengthened and TRC system strengthened specimens. In Table 7, yielding points of the steel rebar are the estimation by measured strains.

It is apparent that the flexural capacity of the RC slab is increased to at least 150% and 200% when the RC slab is TRC strengthened respectively using 1-ply and 2-ply of textile. In view of the behavior from the crack initiation to the steel reinforcement yield, the stiffness of the TRC strengthened specimen increased to 130% on average compared to the unstrengthened specimen. However, the possibility of abrupt failure to occur after the peak load should be a feature to consider carefully in the design of the RC slab strengthened by the TRC system.

### 3.4. Effects of Surface Treatment and No. of Plies of Grids

Figure 13 shows load-displacement curves of specimens with uncoated textile (SN series) and with coated textile (SC series). In the mechanical tests for TRC coupon specimens, the TRC system with coated textile generally developed higher bonding as well as tensile strength relative to the uncoated textile. However, the full-scale flexural test for the slab specimens strengthened with the TRC system indicated that the influence of textile coating on the flexural performance was insignificant regardless of the number of textile plies. All the strengthened specimens tested in this study failed by rupture of the textile, as shown in Figure 14, rather than slip failure or delamination. This result may be due to that fact that the anchoring effect of the textile with in the TRC system.

Figure 13 also shows that the flexural capacity of the specimens strengthened with the TRC system increased as the number of textile plies increased. Figure 13 and Table 7 revealed that the ultimate load-carrying capacity of the SN series (uncoated textile) and SC (coated textile) specimens with 2-ply textile improved to 138% and 128% compared to those with 1-ply textile, respectively. The existing studies indicated that the strengthening effect was not linearly proportional to the amount of textile reinforcement. Although the obtained values have no general significance, Babaeidarabad et al. [12] reported that the peak load when strengthening by 4-ply textile improved by 1.5 times compared to the case where 1-ply textile was used. Schladitz et al. [17] found that, compared to the use of 1 ply, the load-carrying capacity increased by 1.7, 2.1, and 2.5 times when increasing the number of plies by two, three, and four times, respectively.

When an external force is applied to a RC member, internal resistances of materials occur and balance with the external force. Therefore, the maximum load-carrying capacity of the RC member is related to the maximum capacity and the amount of materials. In case of that materials have linear characteristics and tested in one direction like tensile or compressive test, the load is proportional to the amount of materials. Consequently, SN-2 showed higher peak load than SN-1. However, when material have non-linear characteristics and the member is subjected to forces in various directions like flexure and compression, the load-carrying capacity is not proportional to the amount of materials. Due to this phenomenon, the load-carrying capacity of SN-2 was not proportional to the amount of textile grid.

### 3.5. Crack Pattern

Figure 15 and Figure 16 respectively show the bottom and side crack maps of the specimens. All the specimens experienced flexural cracks and failed finally either by flexure failure mode with the rupture of the textile reinforcement or flexure-shear failure mode. The RC specimen and strengthened specimens with 1-ply textile (SN-1 and SC-L1 series) experienced flexural cracking only. On the other hand, the strengthened specimens with 2-ply textile (SN-2 and SC-L2 series) experienced flexural cracking followed by propagation of inclined tensile cracks at approximately 70% of the peak load and finally failed by shear.

Therefore, the test results indicated that the failure mode of the specimens strengthened with the TRC system changed from flexure to flexure-shear mode as the textile reinforcement ratio was increased. Similar failure mode changes were also observed in the tests by other research groups [12,17,18].

### 3.6. Analytical Calculation

The theoretical analysis was based upon the following general assumptions adopted in the analysis of RC structures [32,33]: (1) the strain distribution in the section is linear. In other words, a plane section before flexure remains plane after flexure and is perpendicular to the neutral axis; (2) the strain of concrete around the steel reinforcement and the strain of the steel reinforcement are identical before cracking of concrete or yield of steel; and (3) concrete is weak in tension.

The following assumptions are added for the TRC system: (1) the structure and the TRC system are perfectly bonded and behave monolithically; (2) the resistance to cracking is ignored considering that the TRC matrix is weak in tension; (3) the ultimate compressive strain of concrete is 0.003; (4) the carbon textile grid behaves linearly until rupture (effective tensile strain level in the carbon grid = 0.01 in Table 1); (5) steel behaves bilinearly starting by a linear increase with a slope corresponding to the elastic modulus until the yield strength and followed by a plateau after the yield strength; and (6) the following constitutive laws for concrete apply [34]. Note that the effect of surface coating treatment was not considered in the analytical calculation.
(2)$$f_{c}=f_{c}^{\prime }\left\{\frac{2\varepsilon_{c}}{\varepsilon_{c0}}-{\left(\frac{\varepsilon_{c}}{\varepsilon_{c0}}\right)}^{2}\right\}$$
where $f_{c}$, $\varepsilon_{c}$ = compressive stress and strain levels in concrete; $f_{c}^{\prime }$, $\varepsilon_{c0}$ = maximum stress in flexure of concrete and strain corresponding to the maximum stress of concrete ($1.7f_{c}^{\prime }/E_{c}$); and $\varepsilon_{cu}$, $E_{c}$ = ultimate compressive strain and elastic modulus of concrete.

The equilibrium of the internal forces in the section under a given load can be expressed by Equation (3) where the internal force of concrete can be obtained by Equation (4) [7,21].
(3)$$T_{s}+T_{f}=C_{c}$$
(4)$$C=\alpha_{1}f_{c}^{\prime }\beta_{1}cb$$
where $T_{s}$ = tensile force provided by steel; $T_{f}$ = tensile force provided by TRC; $C_{c}$ = compressive force provided by concrete; c = distance from extreme compression fiber to centroid of steel reinforcement; and b = width of the cross section. The coefficients $\alpha_{1}$ and $\beta_{1}$ can be obtained as follows [7].
(5)$$\alpha_{1}=\frac{3\varepsilon_{c0}\varepsilon_{c}-\varepsilon_{c}^{2}}{3\beta_{1}\varepsilon_{c0}^{2}}$$
(6)$$\beta_{1}=\frac{4\varepsilon_{c0}-\varepsilon_{c}}{6\varepsilon_{c0}-2\varepsilon_{c}}$$

The theoretical peak load of the specimens was computed based on the procedure in [12]. Table 8 compares the analytic and experimental results. The corresponding deflection was calculated from the curvature (=ratio of strain of an element and distance from neutral axis to the element) of the changing section according to the load increase. Since the effect of the eventual coating of the carbon grid cannot be observed, it can be assumed that the SN specimen and the SC series are identical in the theoretical calculation. In Table 8, it appears that the theoretical values obtained using the constitutive laws of each material satisfactorily predict the experimental results. Note that the effect of surface coating treatment was not considered in the analytical calculations.

For practical application because it needs a characteristic bond model which should be studied specially. Therefore, SN and SC specimens show the same result in analytical calculation when the amount of textile grid used is same.

## 4. Field Application

The constructability of the proposed TRC system was examined experimentally by applying the TRC system for the flexural strengthening of a deteriorated RC box culvert. The method adopted in the fabrication of the specimens was applied to construct the TRC system on the concrete structure. In detail, the first mortar layer was laid on the surface of the structure and the textile grid was put in place. The second mortar layer was then laid over the grid. However, if there is a difficulty to put the grid in place at the bottom of the slab then an anchor to fix the textile grid can be used. Accordingly, the grid anchor [35] depicted in Figure 17a,b was developed and applied to effectively install the textile grid on the structure during the construction of the TRC system. The grid anchor consists of a polypropylene (PP) clip and a nail (Figure 17). As shown in Figure 17c, the grid anchor can be installed on the concrete surface to be strengthened with a steel nail for concrete by a gas-powered nailer to a depth of 20~25 mm.

Note that the inner walls of the precast box culvert shown in Figure 18 is generally not critical location for flexure strengthening. However, the inner wall was selected in this study for demonstration purpose. The construction process is illustrated in Figure 18. As shown in Figure 18a, the anchors were installed at a spacing of 200 mm in the vertical and horizontal directions. The inner wall was fully saturated with water prior to apply the TRC system. The shotcrete continued using a shotcrete machine until the designated thickness of the TRC system was obtained (Figure 18b). Finally, the TRC system was finished using a trowel (Figure 18c). This trial application showed that fast construction was possible with remarkable quality of the finished surface. The detailed installation process can be seen in [36].

## 5. Conclusions

This study was carried out to evaluate the flexural strengthening effect provided by the TRC system by executing flexural failure tests on RC slabs strengthened by the TRC system. In addition, a strengthening effect according to the eventual surface treatment of the carbon grid applied in the TRC system as well as an increase in the amount of reinforcement was also observed. The following conclusions can be drawn.

The strengthening of the RC slab by the TRC system made it possible to secure additional load-carrying capacity even after the yield of the steel reinforcement and to expect performance equivalent to that of the RC slab after the occurrence of failure. However, abrupt failure may occur after the yield of steel reinforcement when the TRC-strengthening is applied.

The higher load-carrying capacity and stiffness developed by the TRC-strengthened RC slab resulted in a delay of failure of the structure and reduced the deflection under the same load as compared to the conventional RC slab. For the specimens considered in this study, the load-carrying capacity increased by more than 50% and the stiffness rose by more than 36%.

The results of the present study clearly showed that the surface coated textile grid doubled the bond performance of the TRC coupon specimens. However, the surface coating of textile grids had minimal effect on the load-carrying capacity of the flexural strengthened concrete slab elements. This result may be due to that fact that the anchoring effect of the textile with in the concrete slab elements. Therefore, evaluating the anchorage length of the uncoated textile used in TRC strengthening system under realistic loading conditions should be a major task of future study.

The coating of the surface of the carbon grid used in the TRC system by an abrasive doubled the bond performance but had minimal effect on the load-carrying capacity. This result was attributed to the anchoring length of the textile.

The load-carrying capacity of the RC slab strengthened by the TRC system did not increase proportionally to the adopted amount of carbon grid. The increase of the amount of reinforcement resulted in modification of the failure mode from flexural failure to flexure-shear failure. Consequently, a design check should be done on the failure mode when applying strengthening by the TRC system.

## Figures and Tables

**Figure 1 materials-14-01437-f001:**
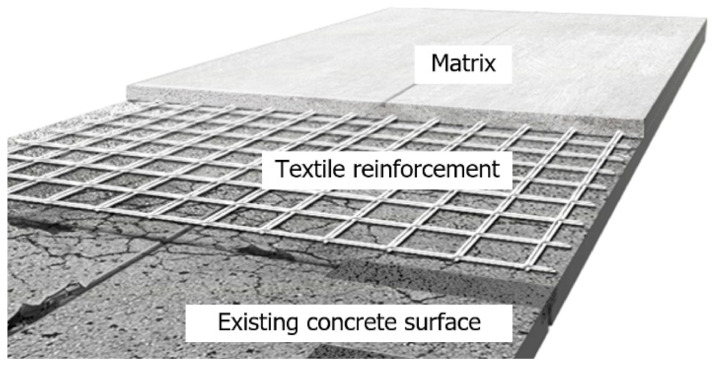
Schematics of TRC system strengthening.

**Figure 2 materials-14-01437-f002:**
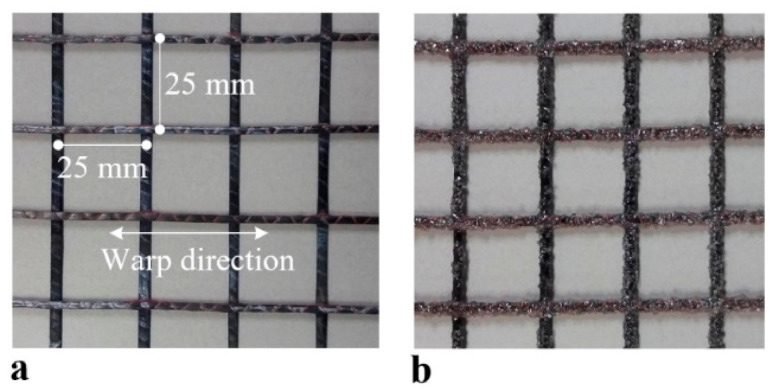
Carbon textile grid: (**a**) uncoated surface; and (**b**) coated surface (Al_2_O_3_ powder #24).

**Figure 3 materials-14-01437-f003:**
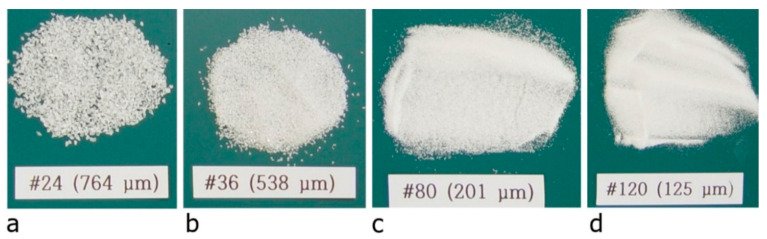
Size of coating materials (white Al_2_O_3_ powder): (**a**) #24; (**b**) #36; (**c**) #80; and (**d**) #120.

**Figure 4 materials-14-01437-f004:**
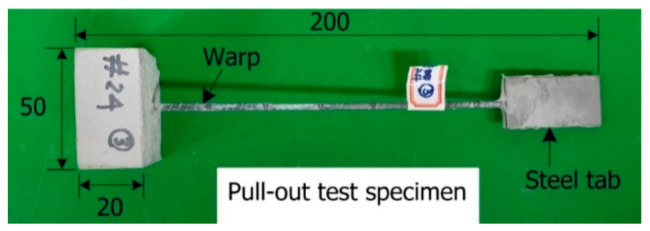
Dimensions of pullout test specimen (unit: mm).

**Figure 5 materials-14-01437-f005:**
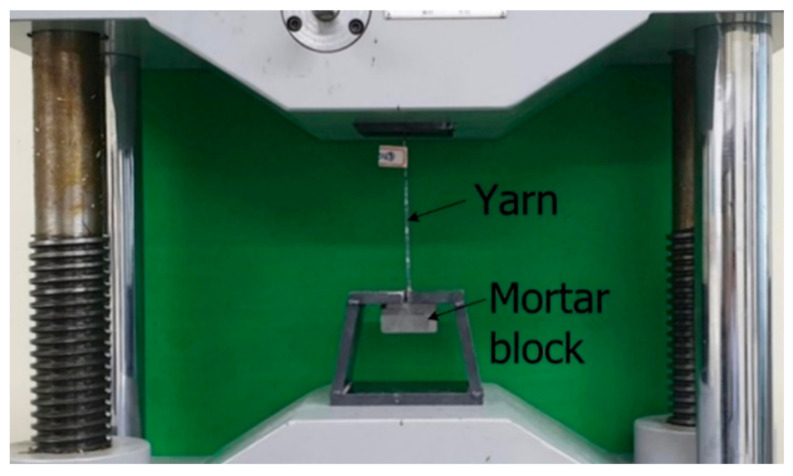
Pullout test setup.

**Figure 6 materials-14-01437-f006:**
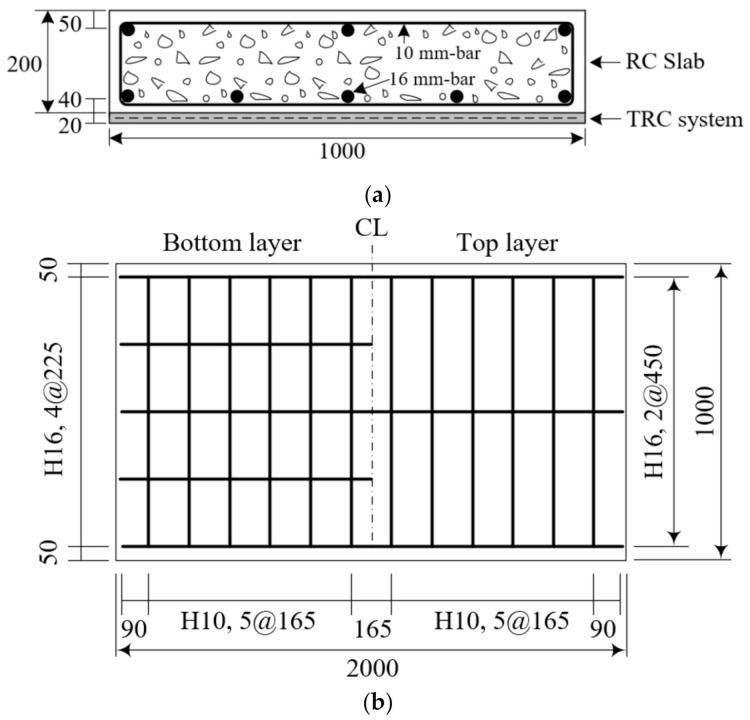
(**a**) Cross-section of a full-scale slab specimen strengthened with TRC system; and (**b**) steel reinforcement details (units: mm).

**Figure 7 materials-14-01437-f007:**
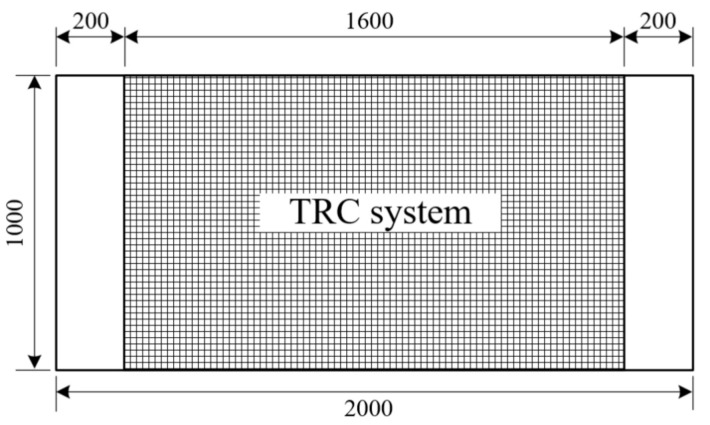
Bottom view of a full-scale slab specimen strengthened with TRC system (units: mm).

**Figure 8 materials-14-01437-f008:**
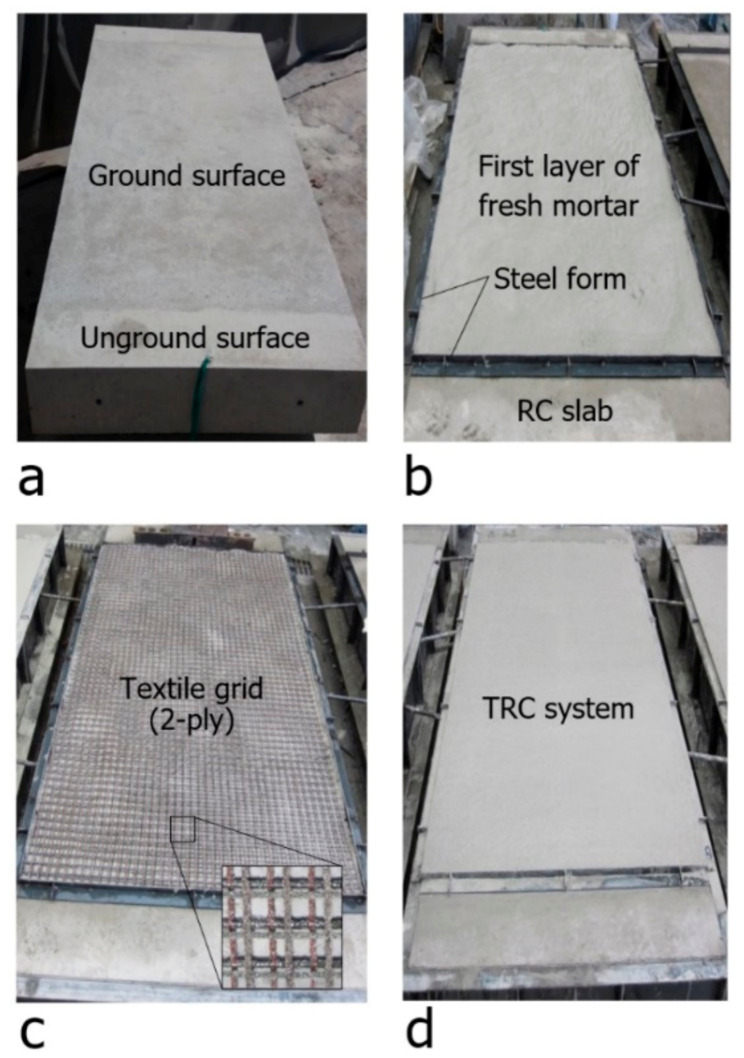
Fabrication process of a full-scale slab specimen: (**a**) ground surface of RC slab; (**b**) 1st mortar layer placement; (**c**) textile grid placement; and (**d**) finished surface of TRC system.

**Figure 9 materials-14-01437-f009:**
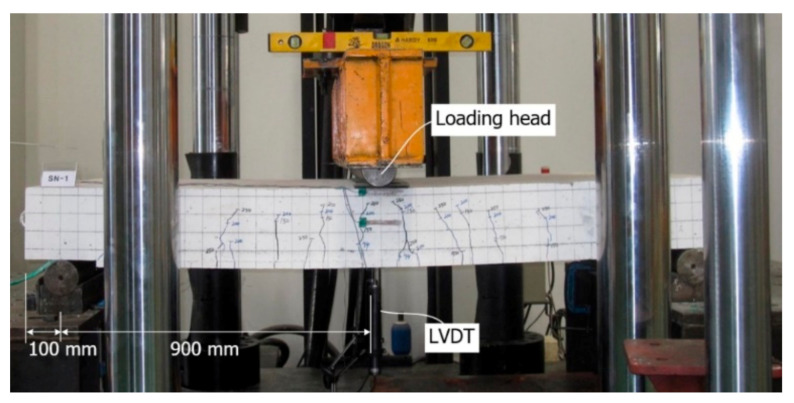
Full-scale flexural test setup and instrumentation.

**Figure 10 materials-14-01437-f010:**
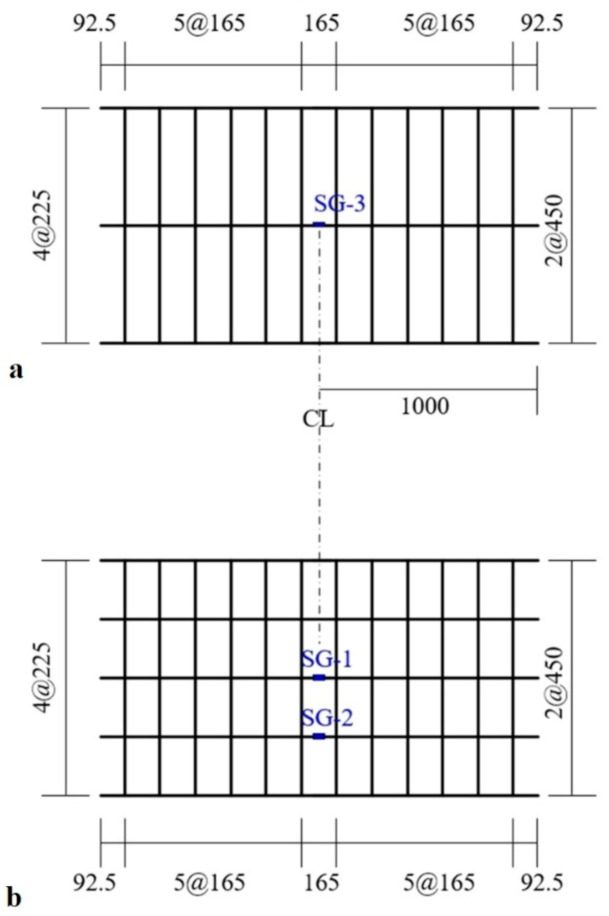
Locations of strain gauges: (**a**) top bars; and (**b**) bottom bars (units: mm).

**Figure 11 materials-14-01437-f011:**
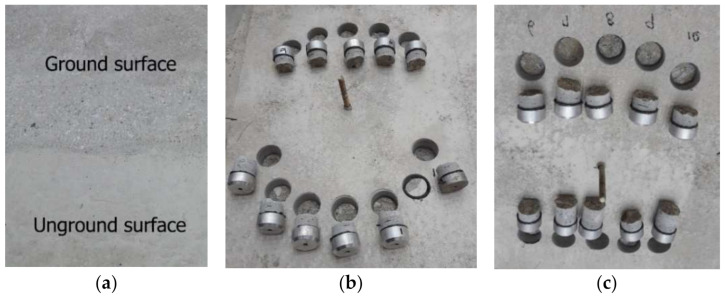
Pull-off test: (**a**) surface conditions before TRC strengthening; (**b**) failure mechanism with unground surface; and (**c**) failure mechanism with ground surface.

**Figure 12 materials-14-01437-f012:**
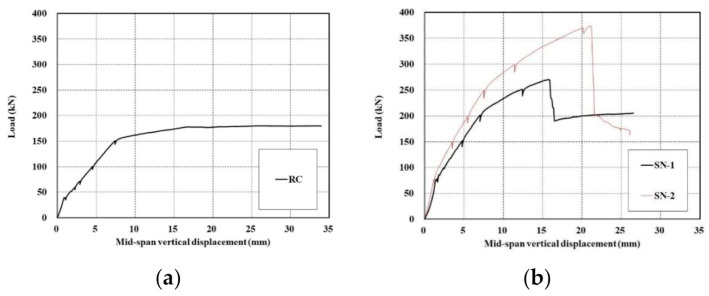
Load-displacement curves of slab specimens: (**a**) control (RC); (**b**) uncoated textile (SN series); (**c**) 1-ply coated textile (SC-L1 series); and (**d**) 2-ply coated textile (SC-L2 series).

**Figure 13 materials-14-01437-f013:**
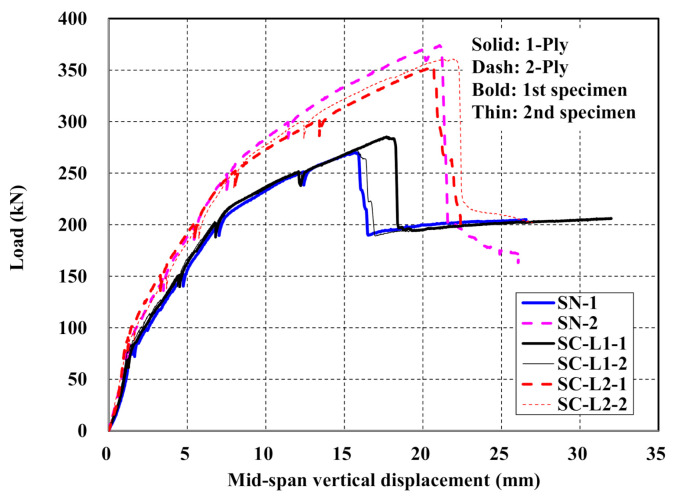
Load-displacement curves of SN and SC series specimens.

**Figure 14 materials-14-01437-f014:**
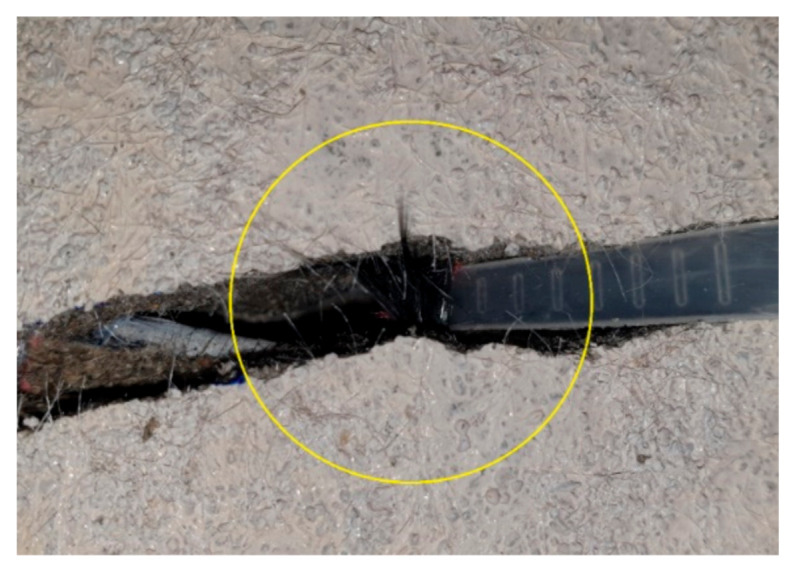
Rupture of textile (SN-1 specimen).

**Figure 15 materials-14-01437-f015:**
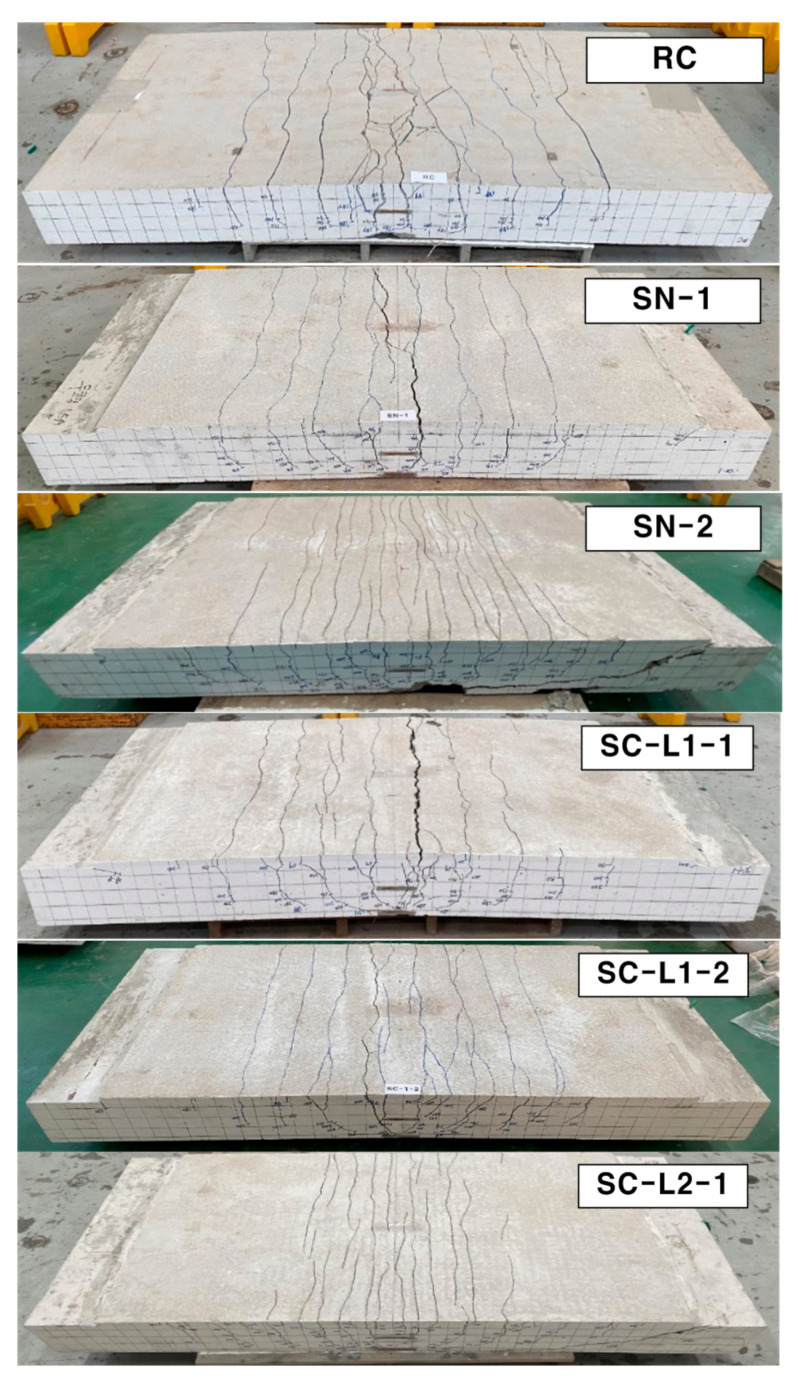
Crack patterns on the bottom side of specimens after failure.

**Figure 16 materials-14-01437-f016:**
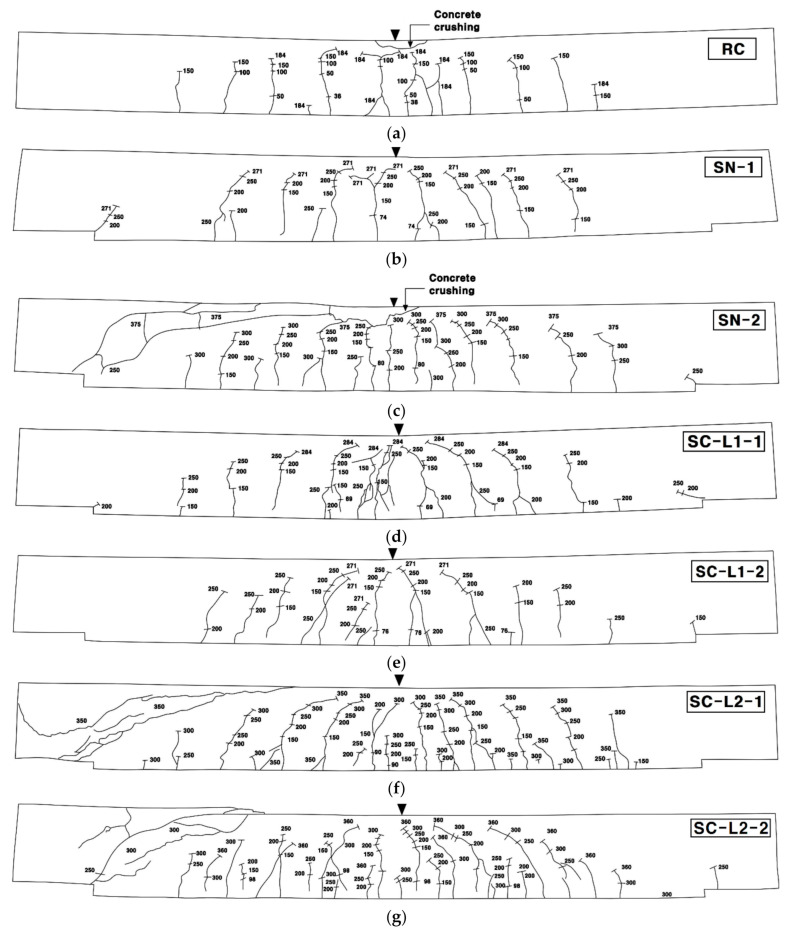
Crack maps on the side of specimens after failure: (**a**) RC; (**b**) SN-1; (**c**) SN-2; (**d**) SC-L1-1; (**e**) SC-L1-2; (**f**) SC-L2-1; and (**g**) SC-L2-2.

**Figure 17 materials-14-01437-f017:**
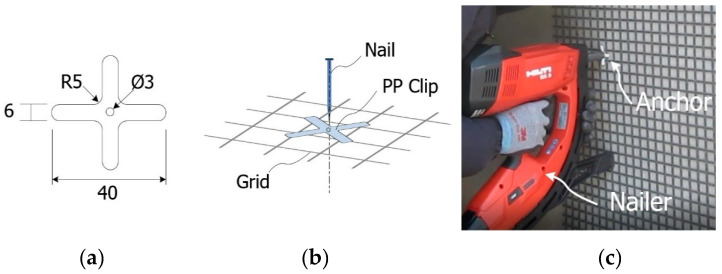
Grid anchor: (**a**) PP clip (unit: mm); (**b**) schematics of grid anchor system; and (**c**) installation example.

**Figure 18 materials-14-01437-f018:**
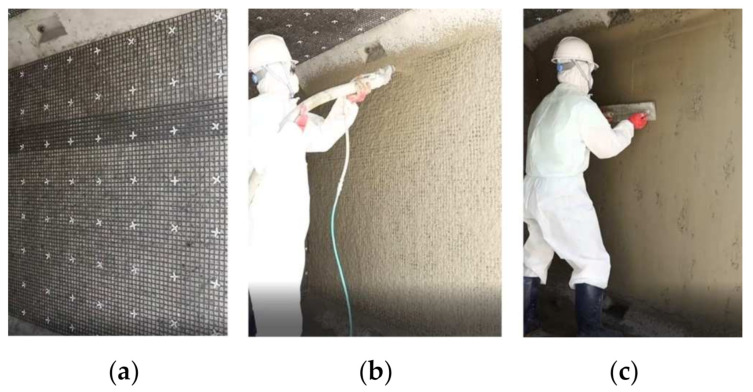
TRC strengthening of a RC culvert: (**a**) installation of grid anchors; (**b**) mortar shotcreting; and (**c**); surface finishing.

**Table 1 materials-14-01437-t001:** Material properties of textile (characteristics values given by the manufacturer, 1 August 2019).

Fiber	Resin	Cross-Sectional Area of Yarn (mm^2^)	Tensile Strength (MPa)	Elastic Modulus (GPa)	Elongation ^1^ (%)
3200 tex	Polystyrene	1.808	1700	200	1.0

^1^ Maximum elongation at failure.

**Table 2 materials-14-01437-t002:** Mixture composition of mortar (unit: kg/m^3^).

Cement ^1^	Admixture	Sand ^2^	Water	Superplasticizer
945.0	77.9	1073.9	483.0	3.2

^1^ Type I Portland cement specified in ASTM C150 [29]. ^2^ Grain size: 0.1~0.4 mm.

**Table 3 materials-14-01437-t003:** Pullout test results (average values of five tests).

Specimen Type	Diameter of Al_2_O_3_ (μm)	Bond Strength (MPa)	CoV
Uncoated	-	4.20	0.041
Coated (#120)	125	8.30	0.053
Coated (#80)	201	8.85	0.050
Coated (#36)	538	8.48	0.033
Coated (#24)	764	8.81	0.026

**Table 4 materials-14-01437-t004:** Characteristics of full-scale slab specimens.

Specimen ID	Textile Coating	No. of Textile Ply	Remarks
RC	-	-	Control, flexural test
PO	Uncoated	1	Pull-off test
SN-1	Uncoated	1	Flexural test
SN-2	Uncoated	2	Flexural test
SC-L1-1	Coated	1	Flexural test
SC-L1-2	Coated	1	Flexural test
SC-L2-1	Coated	2	Flexural test
SC-L2-2	Coated	2	Flexural test

**Table 5 materials-14-01437-t005:** Mixture composition of mortar (unit: kg/m^3^).

Cement ^1^	Granulated Blast Furnace Slag	Sand ^2^	Water	Superplasticizer
466	466	1024	278	7

^1^ Type I Portland cement specified in ASTM C150 [29]. ^2^ Grain size: 0.1~0.4 mm.

**Table 6 materials-14-01437-t006:** Pull-off test results for TRC system.

Surface Condition	Bond Strength (MPa)			CoV
	Maximum	Minimum	Average	
Unground	2.90	1.50	2.16	0.210
Ground	3.00	1.50	2.11	0.191

**Table 7 materials-14-01437-t007:** Flexural failure test results for full-scale slab specimens.

Specimen ID	First Crack			Yield of Steel		Failure		Load Gain (%)
	Load (kN)	Displacement (mm)	Load (kN)		Displacement (mm)	Load (kN)	Displacement (mm)	
RC	39.20	0.87	137.63		6.51	180.45	27.13	100
SN-1	79.68	1.65	202.88		7.27	270.06	15.68	150
SN-2	78.22	1.34	261.16		8.49	374.01	21.36	207
SC-L1-1	69.49	1.27	197.46		6.61	285.25	17.69	158
SC-L1-2	83.51	1.41	218.54		7.76	272.70	15.84	151
SC-L2-1	89.74	1.23	244.92		7.50	353.61	20.67	196
SC-L2-2	96.64	1.63	238.92		7.37	360.78	21.92	200

**Table 8 materials-14-01437-t008:** Comparison of test data with analytical solutions.

Specimen ID	Test		Analysis		Analysis/Test	
	Displacement (mm)	Peak Load (kN)	Displacement (mm)	Peak Load (kN)	Displacement	Peak Load
RC	27.1	180.5	27.6	180.6	1.02	1.00
SN-1	15.7	270.1	-	-	-	-
SC-L1-1	17.7	285.3	-	-	-	-
SC-L1-2	15.8	272.7	-	-	-	-
Average	16.4	276.0	17.9	273.2	1.09	0.99
SN-2	21.4	374.0	-	-	-	-
SC-L2-1	20.7	353.6	-	-	-	-
SC-L2-2	21.9	360.8	-	-	-	-
Average	21.3	362.8	17.9	372.1	0.84	1.03

## Data Availability

The data presented in this study are available on request from the corresponding author.

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
