# Peer review of "Concrete Slab-Type Elements Strengthened with Cast-in-Place Carbon Textile Reinforced Concrete System"

_materials, 2021, doi:10.3390/ma14061437_

Round 1

Reviewer 1 Report

Dear authors,
Thank you for presenting your study! From a purely linguistic point of view, it was very easy to read, and the illustrations are also comprehensible and clearly presented.
In terms of content, I see some room for improvement or the need to explain some statements better or in more detail. This is commented in detail in the enclosed pdf file. It would be nice if you could add one or two sentences each, that should be sufficient. In some places, it is surely enough to change the choice of words a bit.

The most important points are:

  • Literature selection: there is some potential for improvement here in terms of topicality and relevance. 
  • Introduction: too unspecific and partly incorrect. Of course, one can be brief on the topic of TRC in general, but then citations must be appropriately broad and representative. The facts must be verifiable. The very general statements are not always accurate.
  • I miss some information on the material or recommend a partial restructuring within section 2: (A) The information on fine concrete comes too late. A separate subchapter should be formulated for this (e.g. 2.2). Information on the fresh concrete properties would be interesting - these have a decisive influence on the applicability of the TRC to a component. (B) Why was Al2O3 used for the composite improvement? Please complete briefly.
  • I assume that the textile was already impregnated at the factory. Against this background, you should define the word "uncoated" for your paper.
  • Please describe the application process of the additional coating and discuss the reached quality.
  • The description of the measuring equipment is not complete. Later, the strain gauge data are not shown and not discussed. Please complete accordingly.
  • Load increase: please distinguish exactly: "increase something by x %" and "increase something to x %".
  • In several places, the generality of the facts stated/cited must be questioned. In individual cases it is always certainly true, but a generalisation is often not possible. This must be made clear.
  • Section 4: The presented method with the attachment of the textile with additional pins is not practically feasible. In addition, it has often been shown that strengthening also works overhead without additional aids. This must be reflected and discussed in any case.
  • In the summary there are conclusions that have not been discussed before. Accordingly, additions should be made in the preceding sections. A discussion of the procedure presented in section 4 is missing. 
  • Formally, the bibliography needs a little touching up.

I hope the hints are suitable to improve the paper a bit.

Author Response

Theses authors summit the response as attached file.

Reviewer 2 Report

The authors conducted research work on the behaviour of concrete slab specimens strengthened with carbon fibre textile reinforced mortar. Firstly, pull-out test of carbon textile without coating and with different size of aluminum oxide and vinylester resin coating was performed. Secondly, flexural test of slab specimens with variable parameters including the number of carbon textile layers and the presence of coating was performed. Thirdly, field application of textile reinforced mortar to a box culvert was performed. The manuscript is informative and is interesting to the readers. However, the authors shall address and properly rectify the inadequacies: 

- The terminology of textile reinforced concrete should be differentiated from textile reinforced mortar. For the strengthening layer, if it does not contain coarse aggregate, the layer should be called textile reinforced mortar. 

- The reason why the load-displacement curves of specimens SN-1 and SN-2 differ substantially should be explained.

- What was the termination criteria of the flexural test? Were the load-displacement curves trimmed near the end? What was the load-displacement behaviour until the very end of the test?

- The calculation in Table 7 was incomplete. The ratio of analytical displacement to test displacement and the ratio of analytical peak load to test peak load of each specimen should be shown.

- Upon completing the calculation in Table 7, the methodology of accounting for the differences in displacement and peak load among coated and uncoated textile reinforced concrete specimens analytically should be discussed.

- What was the rationale of applying the textile reinforced shotcrete to the interior of a box culvert (as the location is normally not flexure-critical)? More information and justification should be given.

- The citations in the reference list contain inaccuracies and require correction.

Author Response

(The authors gave the same response as above.)

Reviewer 3 Report

1.The abstract must be preceded by 1-2 sentences that introduce the relevance of the research question. Also, you need to add multiple numerical results to the annotation.
2. Al2O3 must be written with subscripts throughout the text.
3. The literature review now looks rather one-sided. It is necessary to add a study of modern positions of structure formation, including taking into account the principles of geomimetics. I recommend, for example, the following articles for review

  1. A D Tolstoy; V S Lesovik; E S Glagolev; A I Krymova. Synergetics of hardening construction systems. IOP Conference Series: Materials Science and Engineering. 2018. 327(3), 032056 . doi: 10.1088/1757-899X/327/3/032056
  2. Haridharan, M.K., Matheswaran, S., Murali, G., Abid, S.R., Fediuk, R., Mugahed Amran, Y.H., Abdelgader, H.S. Impact response of two-layered grouted aggregate fibrous concrete composite under falling mass impact. Construction and Building Materials. Volume 263, (2020), 120628
  3. Youli Lin, Hongjian Du.  Graphene reinforced cement composites: A review. Construction and Building Materials. Volume 265, 30 December 2020, 120312

4. And most importantly, it is not clear from the manuscript what its scientific novelty is. There is a large number of works on textile concrete. Therefore, it is necessary to clearly trace throughout the article, what is the difference from previous works

Author Response

(The authors gave the same response as above.)

Round 2

Reviewer 2 Report

The authors have revised the manuscript to duly address the comments. The revised manuscript is considered acceptable.

Reviewer 3 Report

Good job